# Reconciling Southern Ocean fronts equatorward migration with minor Antarctic ice volume change during Miocene cooling

Suning Hou [1] ✉, Lennert B. Stap [2], Ryan Paul[1], Mei Nelissen [3], Frida S. Hoem [1], Martin Ziegler [1], Appy Sluijs [1], Francesca Sangiorgi [1] & Peter K. Bijl [1]

Gradual climate cooling and $CO_2$ decline in the Miocene were recently shown not to be associated with major ice volume expansion, challenging a fundamental paradigm in the functioning of the Antarctic cryosphere. Here, we explore Miocene ice-ocean-climate interactions by presenting a multi-proxy reconstruction of subtropical front migration, bottom water temperature and global ice volume change, using dinoflagellate cyst biogeography, benthic foraminiferal clumped isotopes from offshore Tasmania. We report an equatorward frontal migration and strengthening, concurrent with surface and deep ocean cooling but absence of ice volume change in the mid–late-Miocene. To reconcile these counterintuitive findings, we argue based on new ice sheet modelling that the Antarctic ice sheet progressively lowered in height while expanding seawards, to maintain a stable volume. This can be achieved with rigorous intervention in model precipitation regimes on Antarctica and ice-induced ocean cooling and requires rethinking the interactions between ice, ocean and climate.

Temperature contrasts between the equator and high latitudes are mitigated through poleward atmospheric and ocean heat transport[1,2]. Variability in the latitudinal sea surface temperature (SST) gradient is mostly a function of polar temperatures, which are much more variable than those at low latitudes because of polar amplification[3]. In turn, polar SSTs, especially offshore Antarctica, vary with prevailing cryosphere conditions, including sea ice extent[4,5]. The steepest part of the latitudinal SST gradient is at mid-latitudes, at the boundary between subtropical gyres and subpolar waters. In the Southern Hemisphere, this is the subtropical front (STF): the northern limit of the Southern Ocean and the Antarctic Circumpolar Current (ACC), and the centre of ocean carbon uptake[6] (Fig. 1). The ACC and associated oceanographic fronts, driven by westerlies and steered by bathymetry[7], regulate deep ocean ventilation[8–10] and heat exchange between low and high latitudes[11,12]. In turn, the latitudinal position of westerlies is influenced by the extent of sea ice around Antarctica[13,14]. Oceanographic

conditions around the ocean fronts thus play a central role in the latitudinal distribution of heat in the Southern Hemisphere, including the heat source that causes basal melt and instability of marine-terminating Antarctic ice sheets[6]. Future projections of polar climate change, and the consequences for cryosphere melt and sea level are highly uncertain[15], because changes in and interactions between Antarctic ice sheets, sea ice and oceanography bear numerous poorly constrained, non-linear feedbacks[6,16]. Important constraints on the functioning of this system in a warming world might come from reconstructions of geologic episodes during which the partial pressure of atmospheric $CO_2$ ($pCO_2$) was as high as projected for the future.

Throughout the Neogene (23–2.58 Ma), $pCO_2$ declined from 800 to 300 parts per million (ppm)[17], global temperatures dropped[18,19], latitudinal SST gradients increased[20] and global ice volume[19,21–23] and sea ice expanded[24]. The current paradigm assigns $pCO_2$ decline as the primary driver, which, through polar amplification of cooling,

[1]Department of Earth Sciences, Utrecht University, Utrecht, The Netherlands. [2]Institute for Marine and Atmospheric research Utrecht, Utrecht University, Utrecht, The Netherlands. [3]NIOZ Royal Netherlands Institute of Sea Research, Texel, The Netherlands. ✉e-mail: s.hou@uu.nl

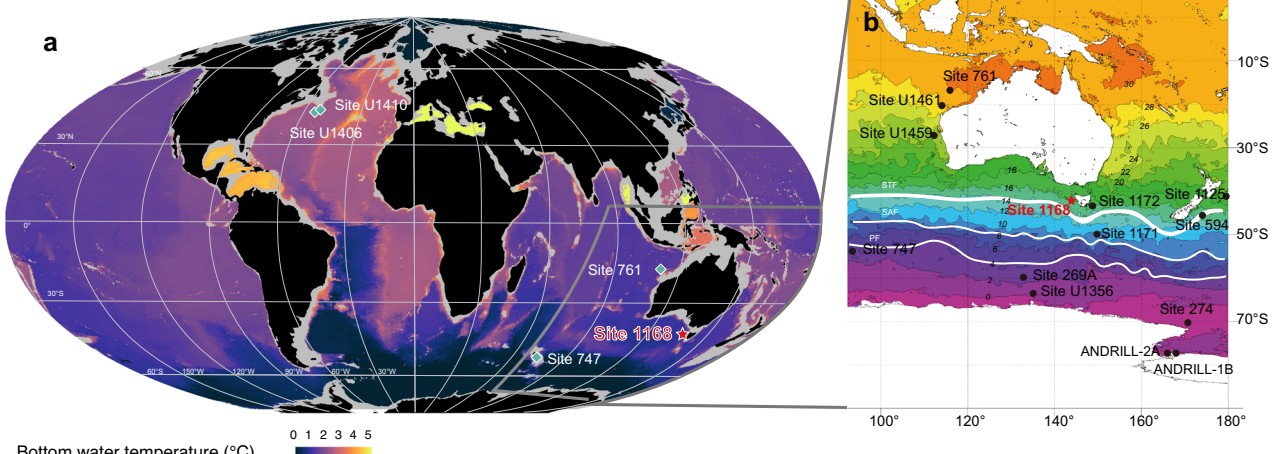

**Fig. 1 | Study area and sites locations. a** Global ocean bottom (>2500 m) water temperature, in blue diamonds the sites from which Miocene clumped isotope data has been generated[36–38], based on refs. [116,117]. Grey line indicates the area of Fig. 1b. **b** Map of the Southern Ocean Sites with modern sea surface temperature[117] and frontal systems positions[118]. STF subtropical front; SAF subantarctic front; PF polar front. Attribution: https://cp.copernicus.org/articles/19/787/2023/ under https://creativecommons.org/licenses/by/4.0/; Illustrations of SAF and PF are added.

stimulates ice growth and cooling in the regions of deep-water formation[18,25,26]. Yet, recent data have challenged this view. A recent study found that Neogene SST gradients increased in the subtropical gyre but decreased from the subtropical front to polar waters[27]. With relatively stable equatorial[28,29] and polar SSTs[30], this indicates that the mid-latitudes, rather than the high-latitudes[27], cooled most profoundly in the Neogene. Antarctic-proximal records suggest a retreated Antarctic ice sheet and warm Antarctic-proximal conditions[24,30,31] during the mid-Miocene Climatic Optimum (MCO) and profound seaward ice sheet advance during subsequent cooling termed the Middle Miocene Climatic Transition (MMCT)[21,23,32–34], in line with $p$CO$_2$ estimates[35]. Along with a rise in deep ocean benthic foraminifera oxygen isotope ratios ($\delta^{18}O_{bf}$), this suggests a strong increase in global ice volume. Yet, the existing series of clumped isotope measurements ($\Delta_{47}$, which deconvolves temperature and ice volume components in $\delta^{18}O_{bf}$ records) on Miocene benthic foraminifera[36–38] suggest higher-than-previously-estimated Bottom Water Temperatures (BWTs) during the MCO, and as a result, large global ice volume. These records also indicate strong BWT cooling during the MMCT, explaining most if not all of the $\delta^{18}O_{bf}$ rise, and therefore little to no ice volume buildup. How this generally connects to far-field sea level changes is still poorly reconciled. However, the uncertainties in clumped isotope data and the limited resolution and temporal range of the records leave ambiguity on the true amount of BWT drop and ice volume buildup during the mid-Miocene.

Like the modern, changes in the Southern Ocean, notably regarding fronts and currents, were likely vital for heat transport towards the ice sheet in the Neogene. A relatively weak ACC, initiated during the Eocene[39], intensified in the late Oligocene ~26 Ma[40] but modern-like strengths only developed in the late Neogene[41]. The development and latitudinal position of the fronts associated to the ACC are, however, still poorly constrained. Meanwhile, a long-term trend of BWT change and how the oceanic processes are coupled to Antarctic ice dynamic is still unclear. To shed light on the links between (Antarctic) ice volume and dynamics, Southern Ocean oceanography and latitudinal SST gradients, we present a detailed reconstruction of Neogene STF migration history and surface and bottom water temperature offshore Tasmania, and pair these with estimates of Antarctic ice volume change from the MCO across the MMCT. We use dinoflagellate cyst (dinocyst) biogeography[42] to reconstruct the position of Southern Ocean currents and fronts and combine these with published SST reconstructions[27]. Finally, from benthic foraminiferal $\Delta_{47}$, we

assess deep-water temperature changes at Ocean Drilling Program (ODP) Site 1168, as well as sea water $\delta^{18}O$ ($\delta^{18}O_{sw}$) as a proxy for Antarctic ice change.

In this study, we demonstrate that there is a strengthening and equatorward migration of the STF from ~53° to ~42° between ~14 Ma and 7 Ma, concurrent with progressive sea surface and bottom water cooling. The deep ocean cooling can completely explain benthic foraminifer $\delta^{18}O$ evolution, implying stable global ice volume. After 7 Ma, the northward shift of the STF is limited by the Australian continent, even though the SSTs continue to decrease. To reconcile expansion of subpolar ocean conditions and progressive Neogene Southern Ocean cooling with stable ice volume and compelling evidence of ice advance, we argue that the Miocene Antarctic ice sheet progressively lowered in height while expanding seawards during the mid-Miocene. We present idealised ice sheet model simulations that physically constrain this hypothesis. This changed geometry induced strong regional oceanographic responses with expansion of sea ice, cooling of the region of bottom-water formation and northwards migration of ocean fronts.

## Results

### Dinoflagellate-based surface oceanographic reconstruction of the subtropical front

The vast majority of the dinocysts encountered in the Neogene sediments from ODP Site 1168 are extant species of the modern Southern Ocean. The use of inferences from modern biogeographic distributions and affinities of dinocyst assemblage clusters[42] (see "Methods" section) hence allows reliable reconstructions of paleoceanographic conditions.

In early Miocene sediments at Site 1168, dinocyst assemblages are dominated by warm/temperate *Spiniferites* spp. (Fig. 2c). This assemblage resembles the Spin-cluster of ref. [42] (Fig. 2b), which now mainly thrives along the northwest coast of Australia and in low latitudes in the eastern Indian Ocean[42]. This cluster is associated with a modern SST of ~29 ± 0.5 °C, a temperature in line with that derived from biomarkers (Fig. 2a). Early Miocene SSTs at Site 1168 were ~13 ± 6 °C (calibration error) warmer than today based on biomarkers, in spite of a ~10 ° more poleward position of the site[43]. Given these SSTs and dinocyst assemblages, we infer a strong influence of the (proto-)Leeuwin Current, delivering heat and sustaining low-latitude dinoflagellate assemblages from western Australia towards the site. It implies that the STF was located to the south of the site. Gradual

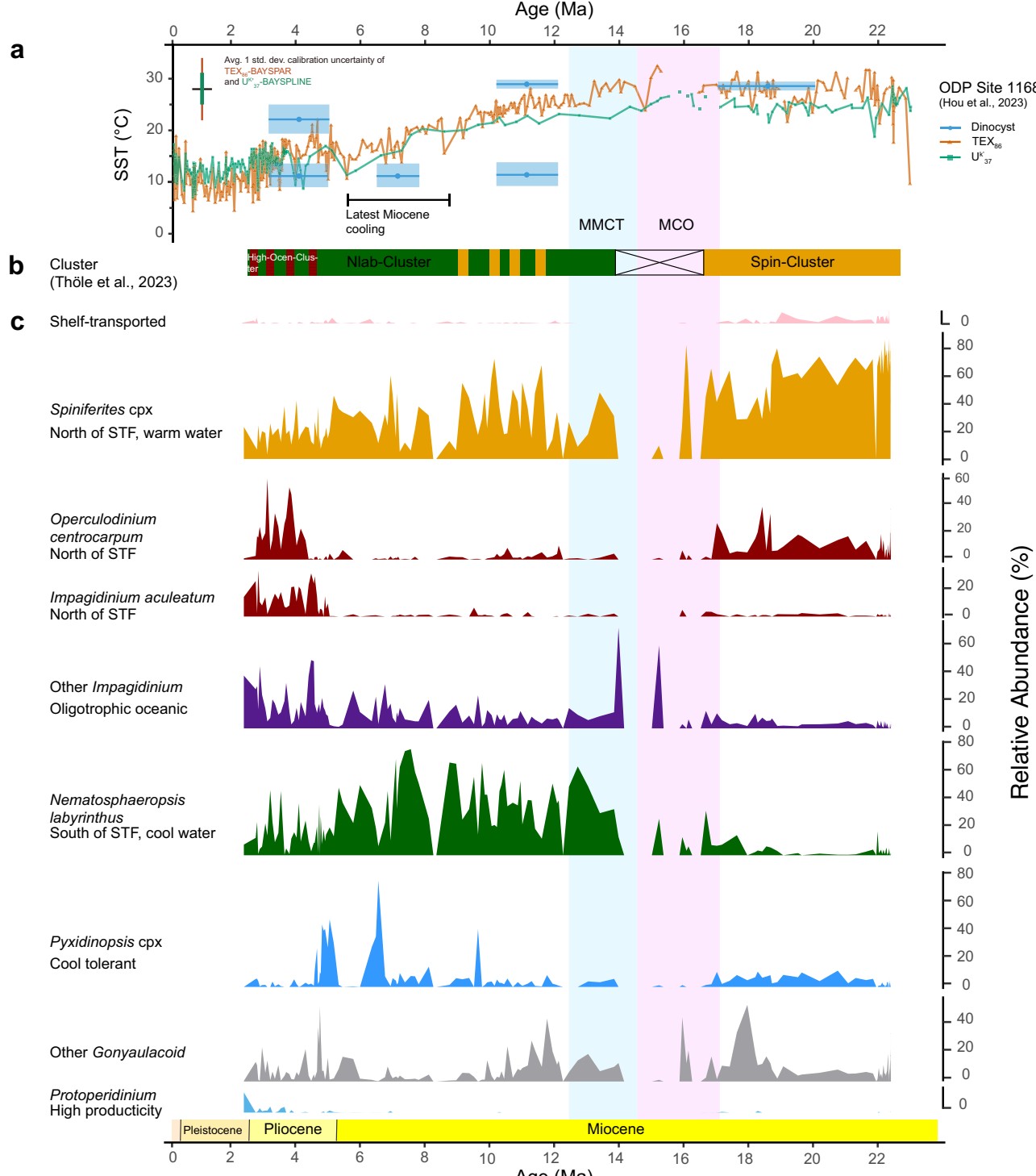

**Fig. 2 | Dinoflagellate cyst assemblage results with published sea surface temperature (SST) records from Ocean Drilling Program Site 1168. a** SST of Site 1168 based on TetraEther indeX of 86 carbons (TEX$_{86}$), alkenone unsaturation ratio (U$^{k}_{37}$)[27], and dinocyst assemblages (this study). TEX$_{86}$, U$^{k}_{37}$ use bayspar[119,120] and bayspline[121] calibrations, respectively. 95% confidence interval is indicated in the panel. Dinocyst-based median SST estimates are based on their environmental affinities[42] in 4 time intervals (see "Methods" section) with 25–75% quantiles. **b** Dinocyst clusters based on ref. 42. **c** Grouped dinocyst assemblages by their ecological affinities based on ref. 42 (see Table S1). Dinocysts are ordered from their known occurrence in latitude from north (top) to south, with uncertain groups and heterotrophic species to the bottom. MCO Mid-Miocene Climatic Optimum, MMCT mid-Miocene Climatic Transition.

increases of *Operculodinium* spp. in this interval suggests gradually cooler-water influence, with an approaching STF from the south. We find occasional northward migrations of the STF (e.g., at ~22 Ma) in sporadic abundance of *N. labyrinthus*, concomitant to SST cooling (Fig. 2c).

Dinocysts are poorly preserved in MCO sediments (Fig. 2c), and Glycerol Dialkyl Glycerol Tetraethers concentrations are low[27], pointing to enhanced sediment oxidation[44]. The available palynological data for the MCO shows that the Spin cluster was replaced by *Impagidinium paradoxum* and *I. patulum*, which in the modern are restricted to

temperate to equatorial open ocean regions between sub-tropical and subpolar systems[45]. Although it is unclear how this dinocyst assemblage differs from the Spin cluster in terms of ocean temperature, the biomarker-based SSTs indicate continued warmth during the MCO at the site (Fig. 2). In any case, Site 1168 remained north of the STF.

The mid Miocene Climatic Transition (MMCT, ~14.5 to 12 Ma) marks the first interval of prevailing *N. labyrinthus* (Fig. 2b, c). This species (Nlab cluster in ref. 42) is found most abundant in sediments south of the STF, in the modern subantarctic zone. We interpret the proliferation of Nlab and a progressive cooling towards subantarctic

zone-like conditions (Fig. 2) as a northward migration of the STF. At MMCT, the STF reached a similar position relative to that of Australia as during the last glacial maximum[46–48] (Fig. 3). Subsequent high-amplitude, short-term fluctuations of dinocyst assemblages between the Nlab- and Spin-cluster, and, albeit less pronounced, biomarker-based SSTs (Fig. 2a), indicate strong (SSTs between 29 °C and 11 °C; Fig. 2c) variability of the latitudinal position of the STF until 7 Ma (Fig. 2).

From ~7 Ma, *N. labyrinthus* started to decrease in abundance. We interpret a southward migration of the frontal systems relative to

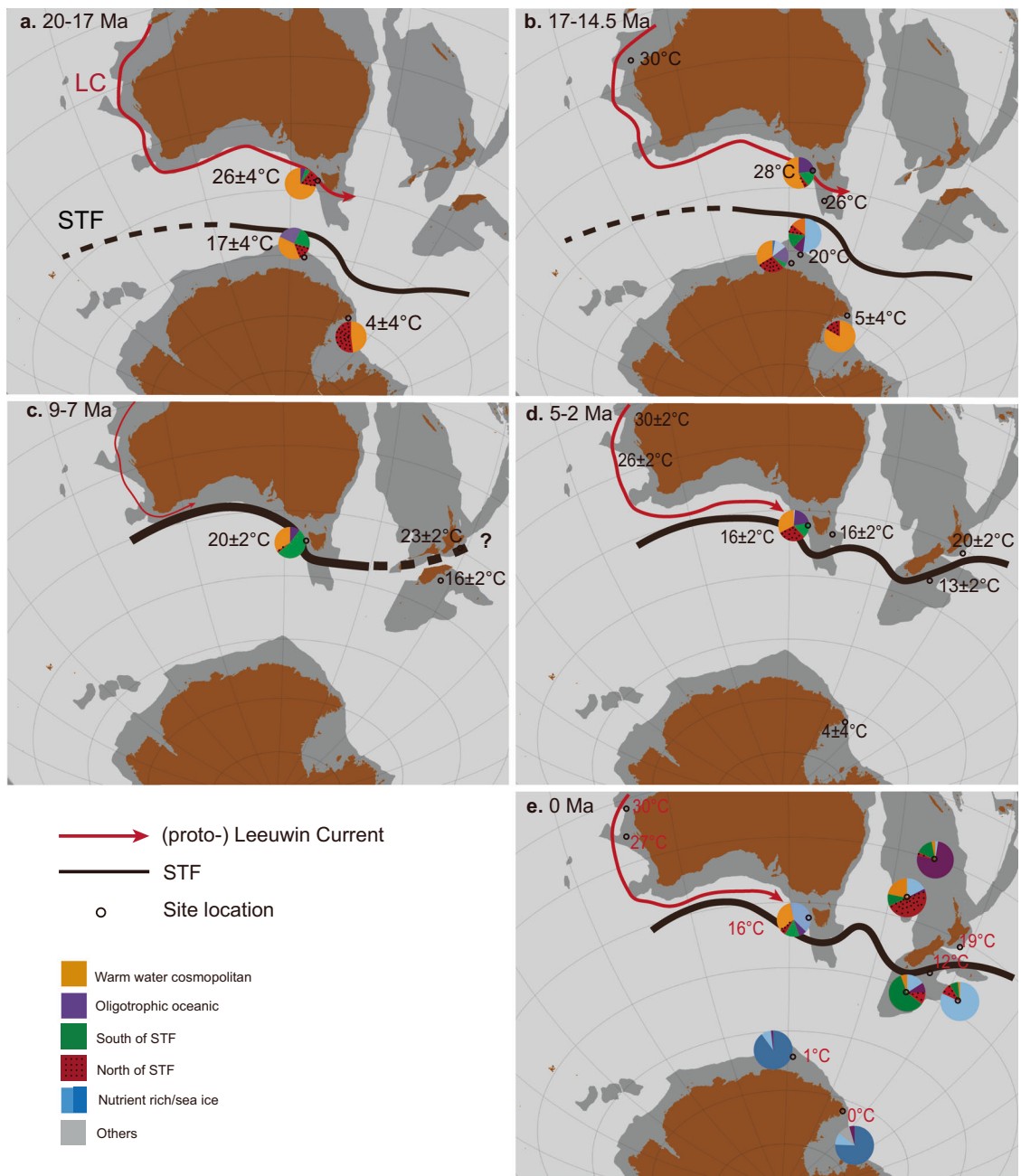

**Fig. 3 | Subtropical Front (STF) migration history in the southeast Indian and southwest Pacific Ocean in 5 time slices from the start of the Neogene. a** early Miocene 20–17 Ma, (**b**) Miocene Climatic Optimum, 17–14.5 Ma (**c**) 9–7 Ma (**d**) 5–2 Ma (**e**) modern, 0 Ma. Average dinocyst assemblages for these time slices at Site 1168 are presented along with those from Site U1356[24], Site 269A[122], Site 274[40] and ANDRILL-2A[78]. The present-day dinocyst distribution is based on Thöle et al.[42]. Red arrow indicates the (proto-) Leeuwin Current. Average sea surface temperature for time slices at Site 1168 are presented along with those from U1356[24], Site 1172[123], Site 1125[20], Site 594[20] and Ross Sea sites[30]. Solid black line indicates the STF, whereby the thickness of the line denotes the relative strength of the STF. Positions of the STF are hypothetically drawn based on this study and refs. 81,118. Paleogeographic position of the continents and sites are generated with the software GPlates[94,124]. Dark brown areas indicate present-day landmass, dark grey indicates continental crust.

Australia from the decline in Nlab and the return of north-of-STF clusters. Apparently, this continued cooling is not directly related to continued shifts of ocean frontal systems, but a cooling of the STF itself and tectonic drift. The Nlab-Cluster and the slightly warmer high-Ocen-Cluster alternate in the Pliocene (4.5–2.5 Ma) on orbital time-scales (Fig. 2b, c), which is close to the modern assemblage, and bracket the modern STF[42] (SSTs between 25 °C and 10 °C; Fig. 2a). The Pliocene stands out as a warmer interval than the end of the Miocene, from both the biomarker-based SSTs (~15 °C)[27] and the dinocysts assemblages. In addition, the Pliocene yields much more abundant *O.centrocarpum* and *I. aculeatum* than the rest of the record, which means that the STF was much closer to ODP Site 1168 in the Pliocene than in the early Miocene, when *Spiniferites* dominated.

Overall, we deduce long-term cooling from the dinocyst assemblages, despite the ~8° northward tectonic movement of the site during the Neogene. There was strong variability over glacial-interglacial climate fluctuations. The STF moved gradually northwards during 22–7 Ma from ~53°S to 42°S (Fig. 3a–c). We infer a concomitant strengthening of the STF from steepened latitudinal SST gradient among mid latitudes[27], and from the fact that the STF was progressively pushed towards the southern margin of the Australian continent. From 7 to 2.5 Ma, the STF moved south from the site again, likely because of Australia's continued northward drift. This allowed for the return of influence of the warm (proto-) Leeuwin Current at the Site (Fig. 3c, d).

### Benthic foraminiferal stable isotope ratios, $\Delta_{47}$ and sea water $\delta^{18}O$

The $\delta^{18}O_{bf}$ and $\delta^{13}C_{bf}$ records generally follow trends recorded at other Southern Ocean sites[36,37] (Fig. 4), including a 1‰ negative offset in $\delta^{18}O$ compared to the CENOGRID compilation[18]. At ~10 Ma, $\delta^{18}O_{bf}$ gradually increases from 1.5‰ (MCO) to 2.5‰, followed by a further rise to ~2.8‰ at the end of the Miocene (~5.3 Ma). Remarkably, we do not record pronounced steps across the MMCT as seen in other records[49]. The pronounced $\delta^{13}C_{bf}$ maxima (from 17 Ma) likely reflects the Monterey carbon isotope excursion[50–52] and values are in line with those in other records.

The benthic foraminiferal clumped isotope data from Site 1168 fill critical mid- and late-Miocene gaps in existing BWT compilations[38] and thus Antarctic ice dynamics (Fig. 5a). BWT, based on $\Delta_{47}$ data in ~1 Myr bins, at Site 1168 decreased gradually from 9.9 ± 4.0 °C (95% confidence interval) in the MCO (17–14.5 Ma) to 5.0 ± 2.5 °C around 10–9 Ma (Supplementary Data 1 and Fig. 5a). While the decreasing trend in mid-Miocene BWT is evident, the confidence intervals on the individual data points leave ambiguity on the significance of the point-to-point cooling. A Student's *t*-test on the bins, however, proves a significant difference in $\Delta_{47}$ between the MCO (17–14.5 Ma) and late Miocene (10–9 Ma; *p* = 0.02; Table S1). Hence, the BWT cooling from the MCO to 9 Ma is significant. The ~8 °C data point at ~8 Ma has only 23 replicates and the longest binned time interval, and because of the resulting high uncertainty we leave this data point out of our interpretations (Fig. S1). By the end of the Miocene (5 Ma), BWTs were slightly elevated (5–6 ± 3 °C) compared to the mid-late Miocene.

Previous studies have pointed out the unexpected warmth of mid-Miocene BWTs in their reconstructions and discussed potential but undiscernible biases on $\Delta_{47}$-based BWT from recrystallisation and pH[36–38,53]. Since benthic foraminifera at Site 1168 are well preserved (Figs. S2 and S3), and seawater chemistry, dissolution and recrystallisation have very limited influence on benthic foraminifera $\Delta_{47}$ composition[54,55], we consider our BWT reconstructions reliable and confirm from Site 1168 the previous inferences of much warmer BWTs in the Miocene than present. By applying a new calibration[56], the BWT shifts by around ~2.4 °C towards lower values (Fig. S4) and results in a smaller global ice volume, but the amplitude of changes remains the same for both BWT and $\delta^{18}O_{sw}$ (see "Methods" section).

## Discussion

The calculated $\delta^{18}O_{sw}$ values from Site 1168 BWTs are 0.3 ± 0.5‰ throughout the MCO and MMCT until 9 Ma (Fig. 5b). The increase in $\delta^{18}O_{bf}$ (~1‰) from 16 to 9 Ma could in principle be all reconciled with the ~5 °C BWT drop we infer from the clumped isotope data (Fig. 5a, b). Thus, our $\Delta_{47}$-based record is different from previous far-field sea-level and deep-sea temperature syntheses based on global $\delta^{18}O$ stack[19,57], one of which recently deconvolved the mid-Miocene $\delta^{18}O_{bf}$ decline

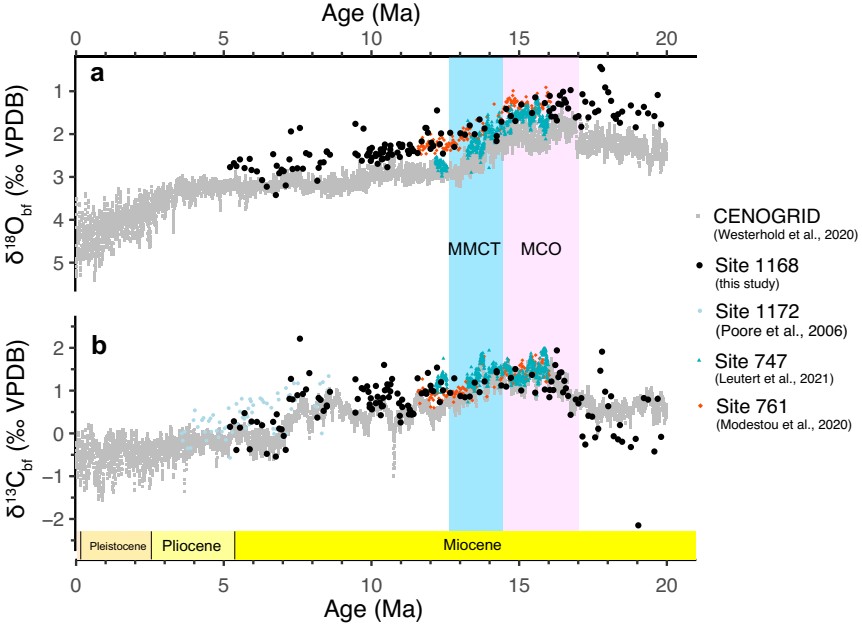

**Fig. 4 | Benthic foraminiferal oxygen and carbon stable isotopes of Site 1168.** **a** oxygen isotopes and (**b**) carbon isotopes of Site 1168 (black dots) together with data from Site 1172[79] (blue dots), Site 747[37] (green triangles), Site 761[36] (orange diamonds) and the CENOGRID stack[18] (grey dots). MCO Mid-Miocene Climatic Optimum (pink shadow), MMCT mid-Miocene Climatic Transition (blue shadow).

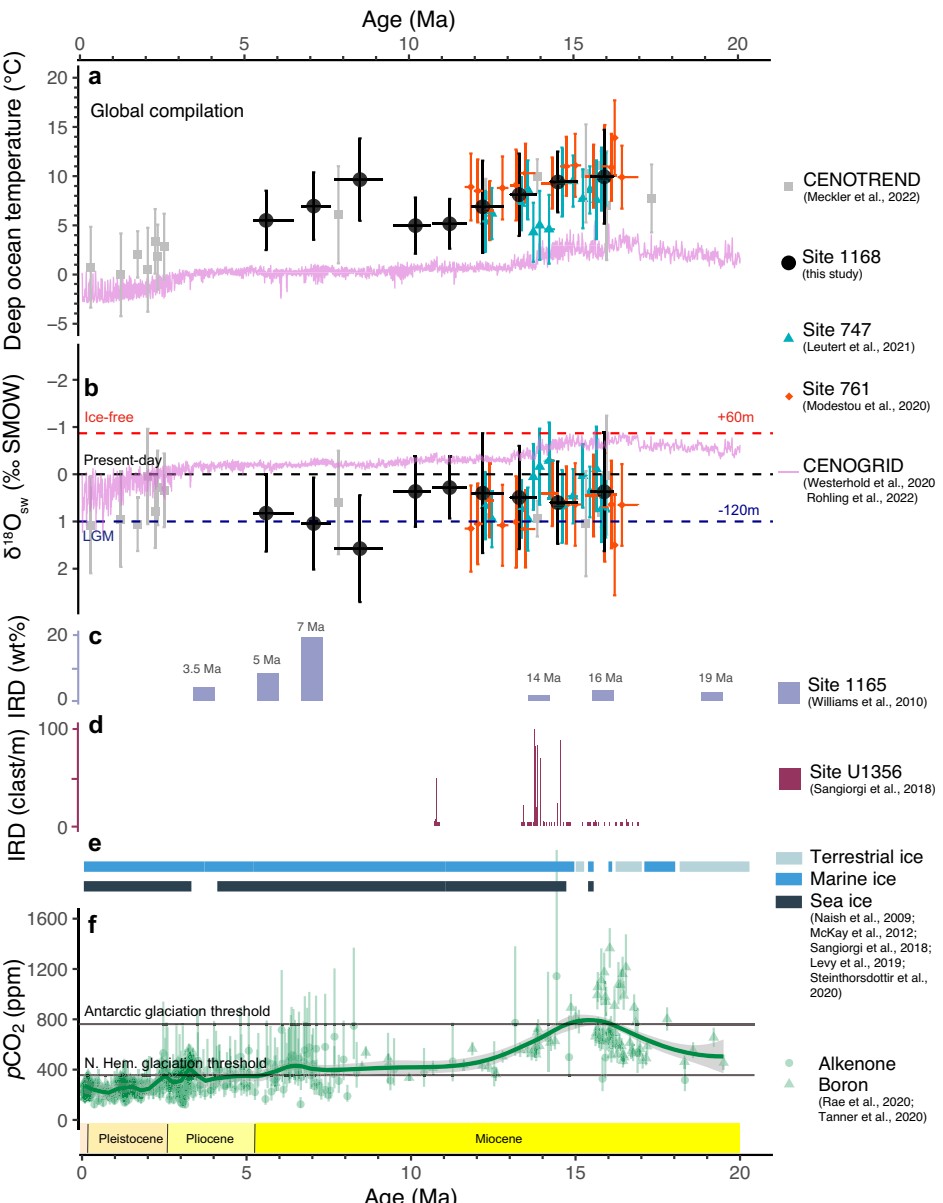

**Fig. 5 | Compilation of records for the Neogene. a** Clumped isotope-based bottom water temperature (BWT) and (**b**) bottom water $\delta^{18}O$ ($\delta^{18}O_{sw}$) of Site 1168 (supplementary data 1) along with data from Site 747 (cyan triangles)[37], Site 761 (orange diamonds)[36] and CENOTREND (grey square)[38]. Horizontal error bars indicate the time interval of each bin. Vertical error bars indicate 95% confidence interval. Violet lines indicate the BWT and $\delta^{18}O_{sw}$ based on Rohling et al.[19]. **c, d** Ice-rafted debris record from Site 1165[87] and U1356[24], units are weight percentage and counts respectively. **e** Qualitative geological record of Antarctic land- (light blue), marine- (blue) and sea ice extent (ink)[21,24,33,34,125,126]. **f** $pCO_2$ reconstructions based on boron isotopes (green triangles) and alkenones $\delta^{13}C$ (green dots)[35,82]. Vertical error bars indicate 95% confidence interval. Solid lines indicate the $pCO_2$ thresholds of glaciation based on DeConto et al.[127].

into in 2.5 °C deep-sea cooling and 25 m of concurrent global average sea-level drop[19]. The discrepancy in $\delta^{18}O_{sw}$ between the study by Rohling et al.[19]. and the clumped isotope data is driven by the difference in absolute BWTs and the magnitude of BWT decline. The uncertainty of the $\Delta_{47}$-based BWT (from 9.9 ± 4 °C to 5 ± 2.5 °C) may allow for some ice volume change (0.3 ± 0.5‰). Given the very similar MCO BWTs derived from multiple sites globally[36,37] we deem the average MCO BWT of 9.9 °C reliable. By binning our MCO data with that of other sites, the uncertainty in that interval can be further reduced to ±1 °C. The 5 ± 2.5 °C BWT in the 10−9 Ma interval is based on most replicates, and thus has the smallest uncertainty. Only when the 10−9 Ma BWT is at the high end of its 95% confidence interval, can the global 25 m RSL (Relative Sea Level) ice volume build-up of Rohling et al.[19] be replicated with our $\Delta_{47}$ data. Apart from $\delta^{18}O$-based

reconstructions, local eustatic reconstructions of the mid and late Miocene are relatively crude and scarce[58,59], and inevitably hindered by local tectonic activities[60,61]. Nevertheless, the inconsistence between $\Delta_{47}$-based absolute $\delta^{18}O_{sw}$ (therefore ice volume) and far-field sea level reconstructions is a general arising question dependent on calibrations[56,62] and requires further exploration[38,63,64]. Given the low probability of that scenario, we conclude that the clumped isotope data imply a stronger cooling and thus less ice volume build-up during MMCT than in the model of Rohling et al.[19].

Although we have confidence in our $\Delta_{47}$-based BWT reconstructions, the higher-than-modern $\delta^{18}O_{sw}$ for the mid-Miocene (and thus a larger than modern global ice volume) seems difficult to reconcile with evidence for Antarctic-proximal sea surface warmth[24,30], Mg/Ca-based deep-sea warmth[65] and high $pCO_2$[35,66,67] during the MCO. The relatively

stable long-term $\delta^{18}O_{sw}$ trend (Fig. 5b) also seems hard to reconcile with major episodes of seaward Antarctic ice expansion across the MMCT, e.g., as suggested by ice-rafted debris[23,24,68]. The only scenario that reconciles all these observations is one whereby a thick AIS was situated inland at the MCO, without marine terminations[69]. Such a high, inland ice sheet would also lead to relatively low oxygen isotope ratios of Antarctic ice[70,71], because the higher-altitude ice sheet would receive less precipitation, and with a lower $\delta^{18}O$[72,73]. Thus, smaller ice volume would be needed for the mass balance if the $\delta^{18}O$ of mid-Miocene land ice was lower than previously assumed. The question is whether such a geometric change in the ice sheet with stable ice volume is dynamically plausible, under realistic boundary conditions. Understanding the detailed interactions between the ocean, climate and ice sheet involved in this situation requires extensive modelling. Here, as a first step, we test the basic viability of a significant change in the volume-to-area ratio of the Miocene Antarctic ice sheet using a stand-alone ice sheet model[60], applying a prescribed precipitation anomaly in conjunction with increased ocean heat ("Methods" section and Fig. 6a, b). This leads to large-scale glaciation at a ~100 ppm higher $CO_2$ level than in the standard setup, yielding a thickened ice sheet interior while the build-up of ice shelves is prohibited and thereby ice area growth impeded (Fig. 6c). Furthermore, from an ice-dynamical perspective, the volume-to-area ratio of the Antarctic ice sheet waxing and waning on orbital timescales is also affected by the forcing amplitude and frequency, because the ice sheet area generally responds faster than volume to climate changes[74]. This implies that a decreased frequency or amplitude around the same mean of forcing variability could lead to an ice sheet that is less extended towards the margins but thicker in the interior, and hence equally voluminous[74].

Following the hypothesis of a dynamic AIS geometry, then, at the MMCT the AIS increased in surface area, advanced seawards, and reduced in height ("Methods" section and Fig. 6, switch from blue to red symbols). When the AIS undergoes spatial expansion, the periphery of the ice sheet receives a greater proportion of precipitation as

compared to the central region. As a consequence of precipitation starvation in the hinterland, the overall elevation of the central AIS reduces. Such a change in geometry would have left global ice volume relatively unaltered but would have had large consequences for ice-ocean interactions and regional climate. Marine-terminating ice sheets provide profound regional cooling to have sea ice expanding[75]. The latitudinal position of westerlies and the sea ice edge determine the position of the STF[7], in absence of continental obstructions[76]. So, in principle, the gradual northwards migration of the STF that we reconstruct is in line with the abundant evidence of seawards land ice expansion across the MMCT[21,23,34]: this induced more marine-terminating ice sheets, and through that a more extensive sea ice. Also on orbital time scales, it is found that marine-terminating ice sheets were strongly sensitive to local solar insolation changes forced by obliquity[34] and so was Southern Ocean paleoceanography[77] (Fig. 5e). This local cooling of high latitudes reduced BWT and pushed ocean fronts northward.

The dinocyst assemblages, combined with previously published SST reconstructions[27] demonstrate the profound latitudinal changes of the STF. In the mid-Miocene, dinocyst assemblages were surprisingly similar between Site 1168 and the Antarctic margin[24,31,78]. Yet, perhaps counterintuitive, the latitudinal SST gradient between Australia[27] and the Ross Sea[30] was largest during MCO. This is because the south Australian Margin was ~10 °C warmer than today, while the inner basins of the Ross Sea remained under local influence of the Antarctic ice sheet and thus relatively cold[30]. In any case, the strong latitudinal SST gradient testifies to the presence of ocean frontal systems that separated mid-latitude water masses from polar water masses. Our reconstructed STF migration is not without corroborating evidence. The southwards STF migration at <7 Ma is coincident with a rapid drop in radiolarian abundance at the East Tasman Plateau[79,80] and decreased K% (Potassium) in southwest Australia[81], both interpreted as a southerly shift in the frontal systems and westerlies relative to the Australian continent. At the same time, at the Agulhas Plateau[82] and in

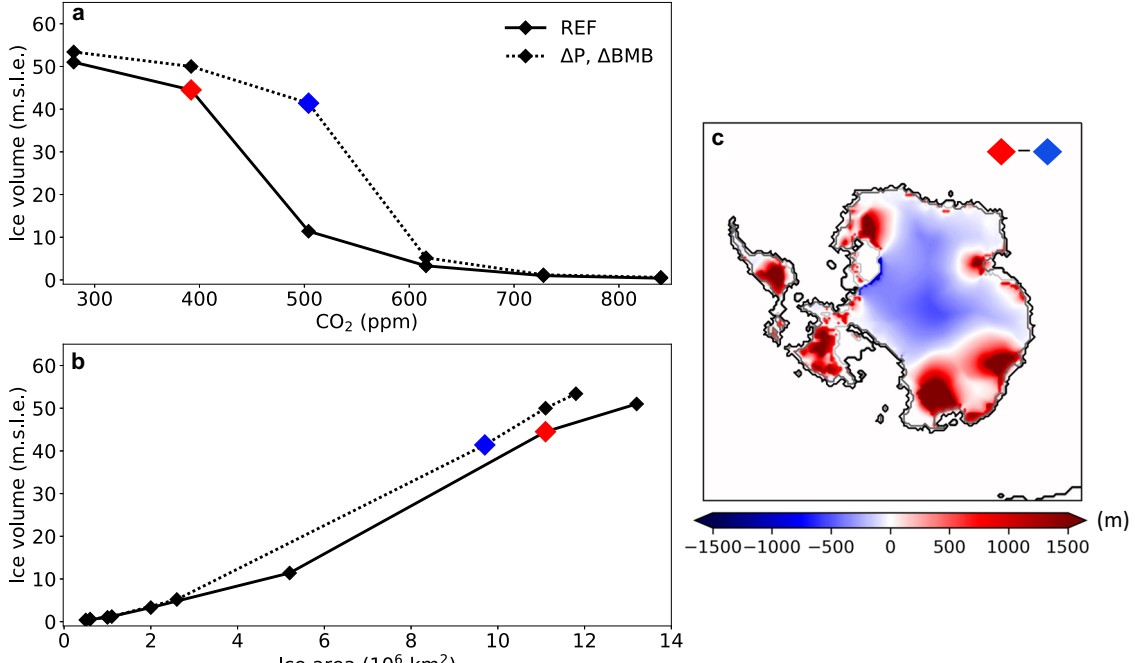

**Fig. 6 | Antarctic ice volumes modelling. a** Simulated equilibrated Antarctic ice volumes at different $CO_2$ levels, and (**b**) the relation between ice volume and ice area, yielded by a 3D thermodynamical ice sheet/shelf moderl (Methods). Results are obtained using the standard climate forcing (solid lines) and applying a fixed precipitation increase and enhanced sub-shelf melt rates (dashed lines). **c** Equilibrated ice thickness difference between the reference simulation at 392 (red) ppm and the simulation with anomalous forcing at 504 ppm (blue). This transition (from the blue to the red symbols) exemplifies our hypothesised Antarctic ice sheet change across the mid-Miocene Climatic Transition.

the South Atlantic[83], oceanographic reconstructions suggest an equatorward migration of oceanic fronts, rather than a southward migration as in Australia. This suggests an asymmetric behaviour of oceanic fronts around Antarctica.

With BWTs around 5 °C at 7–5 Ma, we can attribute the progressive rise in $\delta^{18}O_{bf}$ of 0.2‰ between ~9 and ~6 Ma to about 20 m RSL-equivalent global ice volume build-up (Fig. 5b). This is concurrent with the early significant ice accumulation in Greenland and South America[84,85], and expansion of the west AIS[86], along with enhanced ice-rafted debris off east Antarctica[87] (Fig. 5c, d). The current clumped isotope data compilation (Fig. 5) points to the latest Miocene as the phase of profound global ice volume build-up, rather than the MMCT.

The long-term Southern Ocean BWT cooling signal reconstructed from Site 1168 reflects a high-latitude surface ocean cooling, notably that of the region of deep-water formation. These surface waters where deep-water formed were arguably impacted by the seaward expansion of the ice sheet through katabatic winds[88]. This process expanded sea ice, pushing the westerlies and the STF northward (Fig. 4). In this scenario, the cooling of high latitude surface water and spatial extension of ice reduces the ocean-land thermal contrast and strengthens the polar vortex, leading to less moisture and precipitation transported into Antarctica[89,90]. The progressive cooling of subantarctic waters and increased vertical mixing induced by the northwards-migrated westerlies would have increased the efficiency of the subantarctic ocean carbon sink, the largest single ocean carbon sink system on the planet. As such, the geometric change of the ice sheet could have induced a more efficient ocean carbon storage in the subantarctic zone, which in turn contributed to the lowering of atmospheric $pCO_2$[91] in the Miocene (Fig. 5f).

Taking the above together, the available data show little evidence for Miocene ice volume increase forced by $CO_2$-induced global cooling with polar amplification[29]. First, Neogene surface ocean cooling was not amplified towards the polar regions, as the SST gradient was the largest in the warm MCO and decreasing over the mid-to-late Miocene. Second, the combined STF, BTW and deep ocean $\delta^{18}O$ reconstructions suggest that regional temperatures mostly changed due to geometric changes of the Antarctic ice sheet, rather than the other way around. Northwards expansion of sea ice and subpolar conditions occurred because of advancing marine-terminating ice sheets which induced profound regional cooling. Finally, time intervals with progressive $pCO_2$ decline (MMCT) seem to lack global ice volume increase, while time intervals with relatively stable $pCO_2$ (late Miocene) seem to have profound ice volume growth, suggesting a large role for non-linear feedbacks. These fundamental observations put a perspective on the way radiative forcing and complex feedbacks in ocean-ice-atmosphere interactions shaped Neogene ice volume and global climate trends.

## Methods
### Site description
ODP Site 1168 (42°36.5809′S; 144°24.7620′E; 2463 m modern water depth) (Fig. 1) is located on the continental slope of the west-Tasmanian continental margin, with a modern seafloor temperature of 2.5 °C[92]. The site sits on the northern edge of the Subtropical Convergence zone, which separates warm, saline subtropical waters from comparably cold and fresh subantarctic water masses[93]. During the Neogene, the location of Site 1168 tectonically drifted along with Tasmania and Australia from 52°S at 23 Ma to its modern position at 42 °S[94]. The Neogene bathymetry was lower bathyal/upper abyssal (1000–2500 m), midway on the continental slope[92]. During this northward tectonic drift, the Southern margin of Australia was continuously bathed by the eastward flowing (proto-) Leeuwin Current[40,95]. Hence, Site 1168 is well-suited to study the Neogene evolution of the

STF. We applied the same age model for the sediments as in Hou et al.[27]. (Fig. S5).

### Palynology
We studied 131 samples for palynological content[96]. The processing of sedimentary samples for palynological analysis followed standard procedures at the GeoLab of Utrecht University[97]. Dried sediment samples were crushed and weighed (on average 10 g, standard deviation, SD, of <1 g) before they were dissolved with 30% hydrochloric acid (HCl) and 38% hydrofluoric acid (HF) for carbonate and silicate removal, respectively. The remaining palynological residues were sieved on a 10 μm nylon mesh, using an ultrasonic bath to disintegrate agglutinated organic particles. The palynological residues were mounted on glass slides using glycerine, sealed, and counted (under 200 and 400 magnification) using an Olympus CX41 optical microscope. When possible, at least 200 dinocyst specimens were counted[98]. Samples containing less than (including) 50 dinocyst specimens were excluded for further analysis and interpretation.

We further applied the model of Thöle et al.[42]. (Fig. S6) to infer paleoceanographic conditions from dinocyst assemblages. Specifically, we inferred the 25–75% SST ranges of the clusters in Thöle et al.[42]. that the downcore assemblages compared most to (Fig. 2).

### Foraminiferal preparation
Each sediment sample was freeze-dried, washed over a 63 μm sieve, oven-dried at 50 °C and then dry-sieved into different size fractions. We mainly picked tests of *Cibicidoides mundulus* from the 250–355 μm size fraction for our measurements. We cracked open the picked specimens and ultrasonicated the test fragments in deionized water (3*30 s) to remove adhering sediment, organic lining and nannofossils. The test fragments were dried at room temperature overnight. In order to obtain enough material, other benthic species are also processed. We use *Cibicidoides mundulus* and *Cibicidoides (Planulina) wuellerstorfi* for both stable and clumped isotopes analyses. Data from other benthic or infaunal species *Pyrgo* sp., *Gyroidina soldanii*, *Uvigerina peregerina* are only used for clumped isotopes[99] (Fig. S2).

### Clumped isotope analysis
Clumped isotope measurements were performed using Thermo Scientific MAT 253 and 253 Plus mass spectrometers at the GeoLab of Utrecht University. Both mass spectrometers were coupled to Thermo Fisher Scientific Kiel IV carbonate preparation devices. $CO_2$ gas was extracted from carbonate samples with phosphoric acid at a reaction temperature of 70 °C. A Porapak trap included in each Kiel IV carbonate preparation system was kept at 120 °C to remove organic contaminants from the sample gas. Between each run, the Porapak trap was heated at 120 °C for at least 1 h for cleaning. Every measurement run included a similar number of samples and carbonate standards[100]. In all, 3 carbonate standards (ETH-1, 2, 3) with different $\delta^{13}C$, $\delta^{18}O$ and $\Delta_{47}$ compositions and ordering states were used for monitoring and correction of the results[101]. Two additional reference standards (IAEA-C2 and Merck) were measured in each run to monitor the long-term reproducibility and stability of the instrument. We achieve the necessary precision by averaging ~30 clumped isotope values measured on small (70–100 μg) carbonate samples[101–104]. External reproducibility (1 standard deviation) in $\Delta_{47}$ of IAEA-C2 after correction was 0.033‰. The $\delta^{13}C$ and $\delta^{18}O$ values (reported relative to the VPDB scale) of IAEA-C2 showed an external reproducibility (1 standard deviation) of 0.18‰ and 0.21‰, respectively[105].

### Deep sea temperature and $\delta^{18}O_{sw}$ calculation
We converted the sample $\Delta_{47}$ values (averages over ~30 separate measurements each) into temperature (T, in °C) using a calibration based on various recent datasets from core-top-derived foraminifera,

corrected with the same carbonate standards as used in our study[62]:

$$T = \sqrt{\frac{0.0431 \times 10^6}{\Delta_{47} - 0.1876}} - 273.15 \qquad (1)$$

$\Delta_{47}$-based BWTs were used in combination with $\delta^{18}O_{bf}$ to calculate $\delta^{18}O_{sw}$ (reported relative to VSMOW) with Eq. (9) of Marchitto et al.[106]:

$$\delta^{18}O_{bf}(VPDB) - \delta^{18}O_{sw}(VSMOW) + 0.27 = (-0.245 \pm 0.005) \\ \times T + (0.0011 \pm 0.0002) \times T^2 + (3.58 \pm 0.02) \qquad (2)$$

For these calculations, $\delta^{18}O_{bf}$ values of the genus *Cibicidoides* were averaged over the same intervals as have been used for $\Delta_{47}$ averaging. Calibration uncertainties and measurement error were addressed by applying error propagation. The Meinicke et al.[62,107,108] calibration error was propagated using the R package (clumpedcalib)[109] that utilised a bootstrapped York regression slope-intercept pairs to the bootstrapped mean values for each bin. It should be noted that the calibration error is very small compared to the analytical error. The new calibration[56] excludes benthic foraminifera-based data and it shifts the BWTs parallelly ~2.4 °C colder. As a result, the magnitude of cooling and relative ice volume changes are unaffected (Fig. S4). Since we are using benthic foraminifera as substrates, we decide to keep using the Meinicke calibration.

### Ice sheet modelling

To demonstrate the viability of a precipitation regime change leading to a fundamentally different volume-to-area ratio of the Antarctic ice sheet, we deploy the 3D thermodynamical ice sheet/shelf model IMAU-ICE v1.1.1[110,111]. In the standard set-up[90], climate forcing follows from pre-run warm and cold snapshot climate simulations[112]. The applied climate forcing is transiently calculated based on the prescribed $CO_2$ concentration and the modelled ice sheet size, through a matrix interpolation method[110]. Sea ice is included in the climate model forcing, but a dynamical response is not calculated by IMAU-ICE. Equilibrium experiments are performed at various $CO_2$ levels between preindustrial and 3x preindustrial $CO_2$ values, with insolation at present-day levels and initiated from an ice-free Miocene Antarctic topography[113]. Here, we perform additional sensitivity experiments, in which we apply a fixed precipitation increase and enhanced sub-shelf melt rates. The precipitation anomaly is calculated as 25% of the warm snapshot precipitation fields, sub-shelf melt rates are set to 400 m/yr[114,115].

These sensitivity experiments yield large-scale glaciation at a higher $CO_2$ level (Fig. 6a), and an overall increased volume-to-area ratio (Fig. 6b). Notably, simultaneously reducing the $CO_2$ level from 504 to 394 ppm and removing the anomalous forcing, leads to significantly larger ice sheet area, while the interior ice sheet height is severely reduced (Fig. 6c and S7). These idealised experiments exemplify our hypothesised Antarctic ice sheet change at the MMCT.

### Reporting summary

Further information on research design is available in the Nature Portfolio Reporting Summary linked to this article.

## Data availability

Raw palynological counting, grouped dinocyst data, dinocyst-based SST, BWT bins and stable isotopes data generated in this study have been deposited in Zenodo database: https://doi.org/10.5281/zenodo.8146850. Clumped isotope data generated in this study have been deposited in the EarthChem database: https://doi.org/10.26022/IEDA/112993. The reference simulations analysed in this study are openly accessible from the PANGAEA database: https://doi.org/10.1594/PANGAEA.939114. The additional simulations with increased

precipitation and sub-shelf melt are available from the Zenodo database: https://doi.org/10.5281/zenodo.8308286.

## Code availability

The code for IMAU-ICE v1.1.1-MIO is available from https://github.com/IMAU-paleo/IMAU-ICE/releases/tag/v1.1.1-MIO (last access: 1 September 2023) and https://doi.org/10.5281/zenodo.6352125. The clumped-calib R package is available from https://github.com/japhir/clumpedcalib.

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

## Acknowledgements

We thank Mariska Hoorweg, Natasja Welters, Giovanni Dammers, Desmond Eefting and Arnold van Dijk of the GeoLab for laboratory assistance. We thank IODP and scientists of ODP Leg 189, and technicians at KCC in Kochi, Japan for the help with sampling. We are grateful to Tobias Agterhuis, Ilja Kocken and Elena Domínguez Valdés for insightful discussion regarding clumped isotopes. This research is funded by ERC Starting Grant 802835 to Peter K. Bijl.

## Author contributions

P.K.B. designed the research. S.H., M.N. and F.S.H. processed and analysed samples for palynology. S.H., R.P. and A.S. generated the stable isotopes data. S.H. and R.P. washed the foraminifera samples and generated the clumped isotope data. L.B.S. performed the ice sheet modelling. S.H. wrote the paper with input from P.K.B., A.S., M.Z. and F.S.

## Competing interests

The authors declare no competing interests.
