## [Peer Review File · Nature Communications]

Reconciling Southern Ocean fronts equatorward migration with minor ice volume change during Miocene coolingREVIEWER COMMENTS

Reviewer #1 (Remarks to the Author):

Hou et al NCOMMS 2023

The new manuscript by Hou et al investigates the way in which the Southern Ocean evolved through the Miocene. Specifically, they explore the transient migration of oceanic fronts during the long-term cooling phase from the Miocene Climatic Optimum through the Mid Miocene Climate Transition and into the Late Miocene, using multiple new proxy datasets and analyses. These reconstructions reveal palaeoenvironmental changes that differ substantially from previous work, and so to investigate the drivers of the inferred changes, Hou et al present a series of ice sheet model simulations. Overall, they find that deep water isotopic anomalies during this time, instead of being partitioned 50/50 into temperature and ice volume changes as has previously been suggested, can in fact be almost entirely explained as temperature. This means that ice volume (and by implication sea level) reconstructions need to be reassessed.

Overall I found the paper to be very well written and very well illustrated. It is extremely readable and interesting, the data and methods are logically and objectively presented, and the arguments are clearly explained. I found the study to be remarkably thorough and convincing. In fact, I struggled to find anything of any substance to comment on - my only comments are very minor indeed:

line 32: 'On' should be 'In'

line 35: 'driven by westerlies and bathymetry' - I know what is meant here, but I wonder whether bathymetry really 'drives' ocean currents? Maybe 'steered' or 'deflected' might be better?

Figure 2: It could be useful to highlight along the x axis periods like the MCO and MMCT that are mentioned in the text, just for ease of reference.

line 221: I think 'BTW' should be 'BWT'

line 301: 'enhanced ice-rafted debris off east Antarctica⁸⁹ (Fig. 5c)' - I couldn't find in 5c which of the plotted records this refers to?

line 302: 'this part of the Miocene' - I assume the authors mean the period 6-9 Ma as mentioned earlier in the paragraph - maybe clarify, or use a term like 'Late Miocene' with an indicator on Fig. 2?

line 318: 'Taken the above together'  'Taking...'

All in all an excellent and important piece of work, and one highly deserving of publication in Nature Communications.

N.R. Golledge 7th Aug 2023

Reviewer #2 (Remarks to the Author):

The manuscript "Reconciling equatorward migration of Southern Ocean fronts with minor Antarctic ice volume change during the Miocene cooling" by Hou et al. presents a combination of dinoflagellate and foraminifera-based sea water temperatures with ice sheet modeling in an attempt to solve an apparent controversy in the Miocene climate history around Antarctica focused on the latitudinal migration of the oceanic front systems, specifically the subtropical front (STF). Traditionally, the Mid-Miocene climate transition (MMCT) at ~14 Ma has always been regarded as the glaciation of an important part of Antarctica as expressed in benthic oxygen isotope records. This transition occurred within in a few orbital cycles. The reconstructions presented here though suggest that most of the ice volume change as based on the benthic $\delta^{18}O$ records may rather have been a cooling of deepwater temperatures, using clumped isotopes. This cooling occurred in parallel with a northward moving of

the STF as shown by assemblages of dinoflagellates and their reconstructed sea surface temperatures. The ice sheet modeling indicates that even though ice volume itself may not have changed very much, major changes in the ocean-ice interaction with the resulting oceanic cooling did occur due to changing geometry of the ice sheet.

This study presents a very interesting reconstruction and hypothesis on (global) climate development through the middle-late Miocene, a time period that actually has not received as much attention as the mid-Miocene climate optimum and the MMCT. The combination of different proxy approaches, i.e. dinoflagellate assemblage and SSTs, deep water temperatures based on clumped isotopes, and ice sheet modeling, provides a robust reconstruction with interesting consequences for future climate reconstructions during the Miocene, and possibly even for future climate scenarios.

I do not have major criticisms on this manuscript, but rather some points of interest that may need some clarification or can be included. The manuscript is well-written and very clear to follow and will be a fitting contribution to Nature Communications after minor revisions have been made.

Comments:

Although I am not a modeler, the first and main comment is that it would be very nice to see the effect of the modeling on the actual sea ice extent. Currently the changes are assumed to happen and some proxy studies do seem to support this, but a direct link from the model between the changing geometry and sea ice occurrence would be helpful. And extending this, the logical next step would of course be to see its impact then on the Westerlies and the oceanic fronts.

It would have been nice and supporting the hypothesis to have a reconstruction that is including a high-resolution interval directly at the MMCO, i.e. the time interval where the benthic d18O record is showing the largest changes and always has been interpreted as representing ice volume changes. Whichever signal is in the d18Obenthic record, it did occur within a few orbital cycles, so what was the trigger to make the ice sheet change its geometry, i.e. did this also occur relatively fast, with both a quick response (d18Obenthic) but also keep influencing conditions for the rest of the Miocene?

Although not the main topic of this manuscript, the data is still presented, so I am curious about the increase in *O. centrocarpum* and *I. aculeatum* near the intensification of Northern Hemisphere Glaciation at the end of the Pliocene (Fig. 2). Does this imply a southward migration of the STF again, even though this may be counter-intuitive?

Minor comments/typos:

Line 39: change on to in

Line 79-80: I would rephrase this; it currently reads as if the STF reaches all the way to the bottom (2400 m).

Line 83: MA > Ma

Line 103: remove in

Lines 112-115: check the order of the figures; currently it is Fig. 1, 2, 5, 4, 3.

Line 133: I assume the *N. labyrinthus* equals the *Nematosphaeropsis* panel in figure 2?

Line 165: The Christensen (2021) shows that at some stage in the late Miocene warm subtropical waters may actually have come from the eastside of Australia after Australia moved far enough to the north. The presence of the Leeuwin Current during the Miocene is not completely sure. Some studies indicate that it may not have been formed until the Pliocene related to changes in the Indonesian Throughflow on which it is fully dependent (e.g. Gallagher et al., 2009).

Line 321: BTW > BWT

Sample preparation: the foraminifera were not cleaned following trace metal procedures (e.g. Barker et al., 2003)?

Line 368: Has the use of Pyrgo been tested as reliable for clumped isotopes? It is often included because of its size, but has also been shown as having anomalous geochemical signatures in some settings. Specifically, radiocarbon dating of Pyrgo sp. often gives much older results possibly related to Pyrgo incorporating older, bioturbated material.

Reviewer #3 (Remarks to the Author):

Hou et al. aim to reconstruct the Miocene subtropical front migration and ice-ocean interaction off Tasmania, using a state-of-the-art multi-proxy approach, combining micropaleontology (with dinoflagellate abundance), and isotopic measurements (bottom water temperature with clumped isotope and the global ice volume by combining both clumped and stable isotopes). Thanks to the novel clumped isotope dataset combined with other temperature data, the authors found cooling sea surface and bottom temperatures (SST and BWT, respectively), with evidence of stable ice volume and ice expansion, as confirmed by the model. This is an interesting hypothesis that deserves to be published. The manuscript is well-written with data-driven interpretations and properly referenced. However, I have a few comments.

While reading, I was missing a discussion of SST proxies and their comparison. The authors used 3 different methods and if I understand correctly two were already published (biomarkers - Some references are missing, for example line 77). In the dinoflagellate section, the authors repeatedly mention SST without systematically referring to proxies. I would also suggest writing the uncertainties for all temperatures mentioned in the text.

In addition, I would suggest adding a summary figure with your main results from both isotopic and micropaleontologic data along with the other already published data listed in the discussion such as the IRD. The figure 5 is referred before figures 3 and 4 in the text, emphasizing the importance to present a global figure showing changes in key species/groups of dinoflagellates (STF) and temperatures.

At the end of the abstract, the authors mention conditions specific to their hypothesis related to precipitation patterns and ice-ocean interaction. This sentence is quite broad. I would suggest being more specific.

In the introduction, I would suggest writing a clear paragraph with the objectives. As it stands, the objectives are mixed up in a paragraph (lines 69-80), making it difficult to read them.

In the dinoflagellates part, (lines 146-147), I am not to get how the authors conclude this interpretation, can they clarify?

In the Figure 3: Are black solid lines (STF) based on a model or drawn based on data? If it's based on data and drawing hypothetically, I would suggest using dashed lines, especially for the southwest coast of Australia.

In the method, the Δ^{47} (47) must be in subscript. Also, the authors used a combination of different benthic species for the clumped isotope analyses, including *Pygo* sp., little used in paleoceanography and which can potentially build their shells differently from species classically used in paleoceanography. Have you checked for any differences between the species mix and a mono-species or between classic species and *Pygo* sp.?

Point by point response to all reviewer comments can be found below in **green**.

REVIEWER COMMENTS

Reviewer #1 (Remarks to the Author):

Hou et al NCOMMS 2023

The new manuscript by Hou et al investigates the way in which the Southern Ocean evolved through the Miocene. Specifically, they explore the transient migration of oceanic fronts during the long-term cooling phase from the Miocene Climatic Optimum through the Mid Miocene Climate Transition and into the Late Miocene, using multiple new proxy datasets and analyses. These reconstructions reveal palaeoenvironmental changes that differ substantially from previous work, and so to investigate the drivers of the inferred changes, Hou et al present a series of ice sheet model simulations. Overall, they find that deep water isotopic anomalies during this time, instead of being partitioned 50/50 into temperature and ice volume changes as has previously been suggested, can in fact be almost entirely explained as temperature. This means that ice volume (and by implication sea level) reconstructions need to be reassessed.

Overall I found the paper to be very well written and very well illustrated. It is extremely readable and interesting, the data and methods are logically and objectively presented, and the arguments are clearly explained. I found the study to be remarkably thorough and convincing. In fact, I struggled to find anything of any substance to comment on - my only comments are very minor indeed:

Response to reviewer: We thank the reviewer for the constructive response upon reading our manuscript.

line 32: 'On' should be 'In'

Changes made in the manuscript: corrected.

line 35: 'driven by westerlies and bathymetry' - I know what is meant here, but I wonder whether bathymetry really 'drives' ocean currents? Maybe 'steered' or 'deflected' might be better?

Changes made in the manuscript: rephrased to "and steered by bathymetry."

Figure 2: It could be useful to highlight along the x axis periods like the MCO and MMCT that are mentioned in the text, just for ease of reference.

Changes made in the manuscript: We added 2 bars reflecting the MCO and MMCT time intervals to Figure 2

line 221: I think 'BTW' should be 'BWT'

Changes made in the manuscript: corrected.

line 301: 'enhanced ice-rafted debris off east Antarctica⁸⁹ (Fig. 5c)' - I couldn't find in 5c which of the plotted records this refers to?

Changes made in the manuscript: It was supposed to be Fig. 5d. We have now incorporated IRD visualization into Fig. 5 as suggested by reviewer 3.

line 302: 'this part of the Miocene' - I assume the authors mean the period 6-9 Ma as mentioned earlier in the paragraph - maybe clarify, or use a term like 'Late Miocene' with an indicator on Fig. 2?

Changes made in the manuscript: rephrased to “the latest Miocene”

line 318: 'Taken the above together'  'Taking...'

Changes made in the manuscript: corrected.

All in all an excellent and important piece of work, and one highly deserving of publication in Nature Communications.

Response to reviewer: We thank the reviewer again for the positive assessment.

N.R. Golledge 7th Aug 2023

Reviewer #2 (Remarks to the Author):

The manuscript “Reconciling equatorward migration of Southern Ocean fronts with minor Antarctic ice volume change during the Miocene cooling” by Hou et al. presents a combination of dinoflagellate and foraminifera-based sea water temperatures with ice sheet modeling in an attempt to solve an apparent controversy in the Miocene climate history around Antarctica focused on the latitudinal migration of the oceanic front systems, specifically the subtropical front (STF). Traditionally, the Mid-Miocene climate transition (MMCT) at ~14 Ma has always been regarded as the glaciation of an important part of Antarctica as expressed in benthic oxygen isotope records. This transition occurred within a few orbital cycles. The reconstructions presented here though suggest that most of the ice volume change as based on the benthic $\delta^{18}\text{O}$ records may rather have been a cooling of deepwater temperatures, using clumped isotopes. This cooling occurred in parallel with a northward moving of the STF as shown by assemblages of dinoflagellates and their reconstructed sea surface temperatures. The ice sheet modeling indicates that even though ice volume itself may not have changed very much, major changes in the ocean-ice interaction with the resulting oceanic cooling did occur due to changing geometry of the ice sheet.

This study presents a very interesting reconstruction and hypothesis on (global) climate development through the middle-late Miocene, a time period that actually has not received as much attention as the mid-Miocene climate optimum and the MMCT. The combination of different proxy approaches, i.e. dinoflagellate assemblage and SSTs, deep water temperatures based on clumped isotopes, and ice sheet modeling, provides a robust reconstruction with interesting consequences for future climate reconstructions during the Miocene, and possibly even for future climate scenarios.

I do not have major criticisms on this manuscript, but rather some points of interest that may need some clarification or can be included. The manuscript is well-written and very clear to follow and will be a

fitting contribution to Nature Communications after minor revisions have been made.

Response to reviewer: We thank the reviewer for their positive assessment of the manuscript and its fitness for Nature communications. We respond to the thorough comments in detail below.

Comments:

Although I am not a modeler, the first and main comment is that it would be very nice to see the effect of the modeling on the actual sea ice extent. Currently the changes are assumed to happen and some proxy studies do seem to support this, but a direct link from the model between the changing geometry and sea ice occurrence would be helpful. And extending this, the logical next step would of course be to see its impact then on the Westerlies and the oceanic fronts.

Response to reviewer: Sea ice modelling is indeed of great interest to not only the modelling community but also the paleocommunity. However, the ice sheet model IMAU-ICE that we used in our study can not resolve/simulate sea ice extent. We do not model sea ice with the ice sheet model. There is sea ice in the climate model simulations that is used as forcing, but this can not dynamically respond to the changing ice sheet conditions we simulate.

Changes made in the manuscript: We added “Sea ice extent is not included in the IMAU-ICE model.” in the method, ice sheet modelling section.

It would have been nice and supporting the hypothesis to have a reconstruction that is including a high-resolution interval directly at the MMCO, i.e. the time interval where the benthic $\delta^{18}\text{O}$ record is showing the largest changes and always has been interpreted as representing ice volume changes. Whichever signal is in the $\delta^{18}\text{O}$ benthic record, it did occur within a few orbital cycles, so what was the trigger to make the ice sheet change its geometry, i.e. did this also occur relatively fast, with both a quick response ($\delta^{18}\text{O}$ benthic) but also keep influencing conditions for the rest of the Miocene?

Response to reviewer: We appreciate the point proposed by the reviewer. However, the sampling resolution in this study is not high enough to answer the question, and analysing clumped isotopes in high temporal resolution is complicated because of the necessity to run numerous replicates to achieve low enough uncertainties on the reconstructions. Clearly higher-resolution work is required as follow-up, but this has to be performed on expanded sections where accumulation rates of benthic foraminifer is highest. Previous studies (Modestou et al., 2020; Leutert et al., 2021) have indicated that Δ_{47} -based BWTs were more variable than the benthic $\delta^{18}\text{O}$.

Changes made in the manuscript: No changes made.

Although not the main topic of this manuscript, the data is still presented, so I am curious about the increase in *O. centrocarpum* and *I. aculeatum* near the intensification of Northern Hemisphere Glaciation at the end of the Pliocene (Fig. 2). Does this imply a southward migration of the STF again, even though this may be counter-intuitive?

Response to reviewer: The increase in warm species in the Pliocene is likely due to the position of the Australian continent, which by then approaches its modern position. Meanwhile, there are high amplitude warm/cool species alternations on orbital timescales. We are preparing another manuscript with higher sampling resolution focusing on the late Pliocene, including the M2 and mid-Piacenzian Warm Period, and are working on higher-resolution data for the Pleistocene as well. As the Miocene long-term trends were the focus of this study, we decided not to spend any further explanation on it in this paper.

Changes made in the manuscript: We added a sentence in the results stressing the return of warmth in the Pliocene compared to the late Miocene, but in the interest of later publications, we refrain from going into detail on this further. We added “The Pliocene stands out as a warmer interval than the end of the Miocene, from both the biomarker-based SSTs (~15 °C) (Hou et al., 2023) and the dinocysts assemblages.” in line 218.

Minor comments/typos:

Line 39: change on to in

Changes made in the manuscript: corrected.

Line 79-80: I would rephrase this; it currently reads as if the STF reaches all the way to the bottom (2400 m).

Changes made in the manuscript: rephrased to “...temperature changes at Ocean Drilling Program Site 1168,”

Line 83: MA > Ma

Changes made in the manuscript: corrected.

Line 103: remove in

Changes made in the manuscript: corrected.

Lines 112-115: check the order of the figures; currently it is Fig. 1, 2, 5, 4, 3.

Changes made in the manuscript: corrected.

Line 133: I assume the N. labyrinthus equals the Nematosphaeropsis panel in figure 2?

Changes made in the manuscript: We use “Nematosphaeropsis labyrinthus” in Fig. 2 now.

Line 165: The Christensen (2021) shows that at some stage in the late Miocene warm subtropical waters may actually have come from the eastside of Australia after Australia moved far enough to the north. The presence of the Leeuwin Current during the Miocene is not completely sure. Some studies indicate that it may not have been formed until the Pliocene related to changes in the Indonesian Throughflow on which it is fully dependent (e.g. Gallagher et al., 2009).

Response to reviewer: The Tasman Leakage suggested by Christensen et al. was a subsurface watermass, likely flowing well below the surface but also well above the bottom of our site, therefore it remains a question how much surface warming it caused. The definition of the Leeuwin Current is rather vague, according to Jackson et al. (2019), so it remains somewhat open what one would interpret as Leeuwin-current influence. However, we agree with the reviewer that imposing the terminology of modern ocean currents to paleosettings have a risk of overinterpretation.

Changes made in the manuscript: We use “(proto-) Leeuwin Current” instead throughout the manuscript.

Line 321: BTW > BWT

Changes made in the manuscript: corrected.

Sample preparation: the foraminifera were not cleaned following trace metal procedures (e.g. Barker et al., 2003)?

Response to reviewer: We followed the standard procedure for cleaning foraminifera prior to clumped isotope analyses, which was also used in Agterhuis et al. (2022) and Meckler et al. (2022) for instance. The advantage of this is that results can be easily compared when a consistent methodology was used.

Changes made in the manuscript: No specific changes made.

Line 368: Has the use of Pyrgo been tested as reliable for clumped isotopes? It is often included because of its size, but has also been shown as having anomalous geochemical signatures in some settings. Specifically, radiocarbon dating of Pyrgo sp. often gives much older results possibly related to Pyrgo incorporating older, bioturbated material.

Response to reviewer: We appreciate the concern raised by the reviewer. The conclusions that there are no vital effect offsets of benthic foraminifera for clumped isotopes and there are no discernible species-specific offsets were actually drawn by using Pyrgo sp. (Piasecki et al., 2019).

Changes made in the manuscript: We now cite Piasecki et al. (2019) in the manuscript, method section.

Reviewer #3 (Remarks to the Author):

Hou et al. aim to reconstruct the Miocene subtropical front migration and ice-ocean interaction off Tasmania, using a state-of-the art multi-proxy approach, combining micropaleontology (with dinoflagellate abundance), and isotopic measurements (bottom water temperature with clumped isotope and the global ice volume by combining both clumped and stable isotopes). Thanks to the novel clumped isotope dataset combined with other temperature data, the authors found cooling sea surface and bottom temperatures (SST and BWT, respectively), with evidence of stable ice volume and ice expansion, as confirmed by the model. This is an interesting hypothesis that deserves to be published. The manuscript is well-written with data-driven interpretations and properly referenced. However, I have a few comments.

Response to reviewer: We are grateful to reviewer for the positive feedback and appreciation of our novel hypotheses. We reply on the comments in detail below.

Changes made in the manuscript: no specific changes made.

While reading, I was missing a discussion of SST proxies and their comparison. The authors used 3 different methods and if I understand correctly two were already published (biomarkers - Some references are missing, for example line 77). In the dinoflagellate section, the authors repeatedly mention SST without systematically referring to proxies. I would also suggest writing the uncertainties for all temperatures mentioned in the text.

Response to reviewer: Discussion about the biomarker-based SSTs is indeed relatively missing as they were already published and discussed in Hou et al. (2023), by the same first author.

Changes made in the manuscript: We now cite Hou et al. (2023) in line 78. We also now mention the sources and error of SST where needed.

In addition, I would suggest adding a summary figure with your main results from both isotopic and

micropaleontologic data along with the other already published data listed in the discussion such as the IRD. The figure 5 is referred before figures 3 and 4 in the text, emphasizing the importance to present a global figure showing changes in key species/groups of dinoflagellates (STF) and temperatures.

Response to reviewer: We thank the reviewer for the notice. We intended Fig. 5 to represent the new data generated in this study. But we will incorporate IRD data into this figure.

Changes made in the manuscript: We moved biomarker-based SSTs to Fig. 2 and incorporated previously published IRD data in Fig. 5.

At the end of the abstract, the authors mention conditions specific to their hypothesis related to precipitation patterns and ice-ocean interaction. This sentence is quite broad. I would suggest being more specific.

Changes made in the manuscript: We rephrased to "...precipitation regimes on Antarctica and ice-induced ocean cooling and requires rethinking the interactions between ice, ocean and climate."

In the introduction, I would suggest writing a clear paragraph with the objectives. As it stands, the objectives are mixed up in a paragraph (lines 69-80), making it difficult to read them.

Response to reviewer: We thank the reviewer for the suggestion.

Changes made in the manuscript: We now slightly re-structure the paragraph. "...The development and latitudinal position of the fronts associated to the ACC are, however, still poorly constrained. Meanwhile, a long-term trend of BWT change and how the oceanic processes are coupled to Antarctic ice dynamic is still unclear. To shed light on..."

In the dinoflagellates part, (lines 146-147), I am not to get how the authors conclude this interpretation, can they clarify?

Response to reviewer: During 7–5 Ma, we observed the decline of *N. labyrinthus* and an increase of *Spiniferites* spp., thus, we infer that the STF "moved" southward relative to Australia. Meanwhile, biomarker-based SST dropped. These are likely a combined effect of both surface cooling and tectonic drift.

Changes made in the manuscript: we elaborated more on this, to avoid confusion as follow "... We interpret a southward migration of the frontal systems relative to Australia from the decline in Nlab and the return of north-of STF clusters. Apparently, this continued cooling is not directly related to continued shifts of ocean frontal systems, but a cooling of the STF itself and tectonic drift."

In the Figure 3: Are black solid lines (STF) based on a model or drawn based on data? If it's based on data and drawing hypothetically, I would suggest using dashed lines, especially for the southwest coast of Australia.

Response to reviewer: The solid lines of STF are based on data and drawn hypothetically.

Changes made in the manuscript: We use now dashed lines for the southwest coast of Australia when there are less data supporting. We also added "Positions of the STF are hypothetically drawn based on this study and refs (Orsi et al., 1995; Groeneveld et al., 2017)"

In the method, the Δ_{47} (47) must be in subscript. Also, the authors used a combination of different benthic species for the clumped isotope analyses, including *Pygo* sp., little used in paleoceanography and which can potentially build their shells differently from species classically used in paleoceanography. Have you

checked for any differences between the species mix and a mono-species or between classic species and *Pygo* sp.?

Response to reviewer: Please see the response to the last comment of Reviewer 2

Changes made in the manuscript: We corrected the subscript in line 383 and checked the manuscript throughout for consistency.

REVIEWERS' COMMENTS

Reviewer #2 (Remarks to the Author):

This is a response to the revision of the manuscript by Hou et al. This is my second review of the manuscript after an already very positive review in the first round. The comments and suggestions I made were not all included but still satisfactorily responded to.

Following this I only would like to come back to my comment on the Pliocene increase in *O. centrocarpum*. The authors point out that an upcoming study will go into higher resolution to discuss the Pliocene, and because it is not the main focus of this manuscript, they do not further discuss the increase in *O. cen*. However, the increase is larger than any change in *O. cen* during the Miocene, so it really pulls the eye. So, in that case I would suggest to just leave the Pliocene part out of the dataset; it is only a small part of the record, and would get its own attention in the upcoming manuscript. For the rest I have no further comments and recommend this manuscript for publication in *Nat. Comm.*

Jeroen Groeneveld

Reviewer #3 (Remarks to the Author):

Hou et al. have included all reviewer comments in this revised manuscript. The article is well written and the arguments are well presented, supported by data. I only have one comment regarding the clumped isotope method. A new calibration was released last week (Daëron and Gray, 2023), recalculating the foraminiferal calcification temperatures for the clumped isotope calibration. As a result, the clumped isotope derived temperatures are colder than those obtained with the "old" calibration. Another consequence would be a change in the $\delta^{18}O_{sw}$ reconstruction, with more negative values. I strongly encourage the authors to recalculate the clumped isotopes derived temperature using the new calibration, to check if their hypothesis still holds with the new values of $\delta^{18}O_{sw}$.

Daëron, M., & Gray, W. R. (2023). Revisiting oxygen-18 and clumped isotopes in planktic and benthic foraminifera. *Paleoceanography and Paleoclimatology*, e2023PA004660.

Point by point response to reviewers are shown below in blue.

REVIEWERS' COMMENTS

Reviewer #2 (Remarks to the Author):

This a response to the revision of the manuscript by Hou et al. This is my second review of the manuscript after an already very positive review in the first round. The comments and suggestions I made were not all included but still satisfactorily responded to.

Following this I only would like to come back to my comment on the Pliocene increase in *O. centrocarpum*. The authors point out that an upcoming study will go into higher resolution to discuss the Pliocene, and because it is not the main focus of this manuscript, they do not further discuss the increase in *Ocen*. However, the increase is larger than any change in *Ocen* during the Miocene, so it really pulls the eye. So, in that case I would suggest to just leave the Pliocene part out of the dataset; it is only a small part of the record, and would get its own attention in the upcoming manuscript.

For the rest I have no further comments and recommend this manuscript for publication in Nat. Comm.

Jeroen Groeneveld

Response to the reviewer: We thank the reviewer for the positive comments on our manuscript. The upcoming Pliocene manuscript focuses on high resolution for the interval 2.9–3.5 Ma, so we do agree that the long-term changes in Pliocene dinocysts assemblages fits this manuscript.

Changes made in the manuscript: We revised Figure 2 to better illustrate the Nlab/High-*Ocen* alternations. We added more context in the main text: "In addition, the Pliocene yields much more abundant *O.centrocarpum* and *I. aculeatum* than the rest of the record, which means that the STF was much closer to ODP Site 1168 in the Pliocene than in the early Miocene, when *Spiniferites* dominated." in line 156.

Reviewer #3 (Remarks to the Author):

Hou et al. have included all reviewer comments in this revised manuscript. The article is well written and the arguments are well presented, supported by data. I only have one comment regarding the clumped isotope method. A new calibration was released last week (Daeron and Gray, 2023), recalculating the foraminiferal calcification temperatures for the clumped isotope calibration. As a result, the clumped isotope derived temperatures are colder than those obtained with the "old" calibration. Another consequence would be a change in the $\delta^{18}\text{O}_{\text{sw}}$ reconstruction, with more negative values. I strongly encourage the authors to recalculate the clumped isotopes derived

temperature using the new calibration, to check if their hypothesis still holds with the new values of $\delta^{18}\text{O}_{\text{sw}}$.

Daëron, M., & Gray, W. R. (2023). Revisiting oxygen-18 and clumped isotopes in planktic and benthic foraminifera. *Paleoceanography and Paleoclimatology*, e2023PA004660.

Response to the reviewer: We thank the reviewer for the positive comments on our revised manuscript. Applying the calibration (equation 5 in Daëron and Gray, 2023) to our data, the BWT shift by around $\sim 2.4^\circ\text{C}$ towards lower values (see the figure below), but are within the 95% confidence intervals of our originally reported values. Since the overall trends in the dataset remain unaffected our main conclusions regarding BWT changes still hold. As the reviewer points out correctly, the reconstructed $\delta^{18}\text{O}_{\text{sw}}$ values will do shift towards more negative values as well when using the new calibration and this would in turn point to a smaller global ice volume during the time interval of interest. However, determination of the absolute ice volume is not subject of our manuscript as we instead focus our discussion on relative ice volume changes, that are unaffected by the calibration we choose. For the main part of the manuscript, we keep the calibration of Meinicke et al. that we also applied in our original submission. Previously published clumped BWT dataset that we use for comparison in our manuscript use the same calibration. Another reason is that the calibration developed by Daëron and Gray (2023) excludes benthic foraminiferal based data.

Changes made in the manuscript: We added the comparison figure below to the supplementary information and made comments on the Daëron and Gray (2023) calibration in the main text.

We added to the main text “By applying a new calibration (Daëron and Gray, 2023), the BWT shifts by around $\sim 2.4^\circ\text{C}$ towards lower values (Fig. S4) and results in a smaller global ice volume, but the amplitude of changes remains the same for both BWT and $\delta^{18}\text{O}_{\text{sw}}$.” in line 227;

“An alternative calibration will only influence the absolute values of BWT and $\delta^{18}\text{O}_{\text{sw}}$. The magnitude of cooling and relative ice volume changes are unaffected” in line 440.

We rephrased line 257–259 as “Nevertheless, the inconsistency between $\Delta 47$ -based **absolute** $\delta^{18}\text{O}_{\text{sw}}$ (therefore ice volume) and far-field sea level reconstructions is a general arising question **dependent on calibrations**^{67,78} and requires further exploration^{38,79,80}.”